# SoftPatch: Unsupervised Anomaly Detection with Noisy Data

**Xi Jiang**[1]
jiangx2020@mail.sustech.edu.cn

**Jianlin Liu**[2]
jenningsliu@tencent.com

**Jinbao Wang**[1*]
wangjb@sustech.edu.cn

**Qian Nie**[2]
stephennie@tencent.com

**Kai Wu**[2]
lloydwu@tencent.com

**Yong Liu**[2]
choasliu@tencent.com

**Chengjie Wang**[2]
jasoncjwang@tencent.com

**Feng Zheng**[1*]
zhengf@sustech.edu.cn

[1]Southern University of Science and Technology,
Department of Computer Science and Engineering
[2]Tencent, Youtu Lab

## Abstract

Although mainstream unsupervised anomaly detection (AD) algorithms perform well in academic datasets, their performance is limited in practical application due to the ideal experimental setting of clean training data. Training with noisy data is an inevitable problem in real-world anomaly detection but is seldom discussed. This paper considers label-level noise in image sensory anomaly detection for the first time. To solve this problem, we proposed a memory-based unsupervised AD method, SoftPatch, which efficiently denoises the data at the patch level. Noise discriminators are utilized to generate outlier scores for patch-level noise elimination before coreset construction. The scores are then stored in the memory bank to soften the anomaly detection boundary. Compared with existing methods, SoftPatch maintains a strong modeling ability of normal data and alleviates the overconfidence problem in coreset. Comprehensive experiments in various noise scenes demonstrate that SoftPatch outperforms the state-of-the-art AD methods on the MVTecAD and BTAD benchmarks and is comparable to those methods under the setting without noise.

## 1 Introduction

Detecting anomalies by only nominal images without annotation is an appealing topic, especially in industrial applications where defects can be extremely tiny and hard to collect. Unsupervised sensory anomaly detection, also called covariate shift detection [1; 2], is proposed to solve this problem and has been largely explored. Recent deep learning methods [3; 4; 5; 6; 7] usually model the AD problem as a one-class learning problem and employ computer visual tricks to improve the perception where a clean nominal training set is provided to extract representative features. Most previous unsupervised AD methods have to measure the distance between the test sample and the standard dataset distribution to determine whether a sample differs from the standard dataset. Even though recent methods have achieved excellent performance, they all rely on the clean training set to extract

---

*Corresponding Author

36th Conference on Neural Information Processing Systems (NeurIPS 2022).

nominal features for later comparison with anomalous features. Putting too much faith in training data can lead to pitfalls. If the standard normal dataset is polluted with noisy data, i.e., the defective samples, the estimated boundary will be unreliable, and the classification for abnormal data will have low accuracy. In general, current unsupervised AD methods are not designed for and are not robust to noisy data.

However, in real-world practice, it is inevitable that there are noises that sneak into the standard normal dataset, especially for industrial manufacturing, where a large number of products are produced daily. This noise usually comes from the inherent data shift or human misjudgment. Meanwhile, existing unsupervised AD methods [8; 9; 10] are susceptible to noisy data due to their exhaustive strategy to model the training set. As in Fig. 1, noisy samples easily misinform those overconfident AD algorithms, so algorithms misclassify similar anomaly samples in the test set and generate wrong locations. Additionally, AD with noisy data can be developed to a fully unsupervised setting, which discards the implicit supervised signal that the training set is all defect-free, compared with the previous unsupervised setting in AD. This setting helps to expand more industrial quality inspection scenarios, i.e., rapid deployment to new production lines without data filtration.

In this paper, we first point out the significance of studying noisy data problems in AD and especially in unsupervised sensory AD. Our solution is inspired by one of the recent state-of-the-art methods, PatchCore [8]. PatchCore proposed a method to subsample the original CNN features of the standard normal dataset with the nearest searching and establish a smaller coreset as a memory bank. However, the coreset selection and classification process are vulnerable to polluted data. In this regard, we propose a patch-level selection strategy to wipe off the noisy image patch of noisy samples. Compared to conventional sample-level denoising, the abnormal patches are separated, and the normal patches of a noise sample are exploited in coreset. Specifically, the denoising algorithm assigns an outlier factor to each patch to be selected into coreset. Based on the patch-level denoising, we propose a novel AD algorithm with better noise robustness named SoftPatch. Considering noisy samples are hard to be removed completely, SoftPatch utilizes the outlier factor to re-weight the coreset examples. Patch-level denoising and re-weighting the coreset samples are proved effective in revising misaligned knowledge and alleviating the overconfidence of coreset in inference. Extensive experiments in various noise scenes demonstrate that SoftPatch outperforms the state-of-the-art (SOTA) AD methods on MVTec Anomaly Detection (MVTecAD) [11] benchmark. Meanwhile, due to the noise in existing datasets, SoftPatch achieves optimal results on the original BTAD [12] dataset. The code can be found in `https://github.com/TencentYoutuResearch/AnomalyDetection-SoftPatch`.

Our main contributions are summarized as follows:

- To the best of our knowledge, we are the first to focus on the image sensory anomaly detection with noisy data, which is a more practical setting but seldom investigated. Existing image sensory AD methods fully trust the training set's cleanliness, leading to their performance degradation in noise interference.

- We propose a patch-level denoising strategy for coreset memory bank, which essentially improves the data usage rate compared to conventional sample-level denoising. Based on this strategy, we apply three noise discriminators which strengthen model robustness by combining the re-weighting of coreset.

- We set a baseline for unsupervised AD with noisy data, which performs well in the settings with additional noisy data and the general settings without noise, providing a new view for further research.

## 2   Related Work

### 2.1   Unsupervised Anomaly Detection

**Training with agent tasks.** Also known as self-supervised learning, agent tasks is a viable solution when there is no category and shape information of anomalies. Sheynin et al. [13] employ transformations such as horizontal flip, shift, rotation, and gray-scale change after a multi-scale generative model to enhance the representation learning. Li et al. [14] mention that naively applying existing self-supervised tasks is sub-optimal for detecting local defects and propose a novelty agent task named CutPaste, which simulates an abnormal sample by clipping a patch of a standard image and pasting

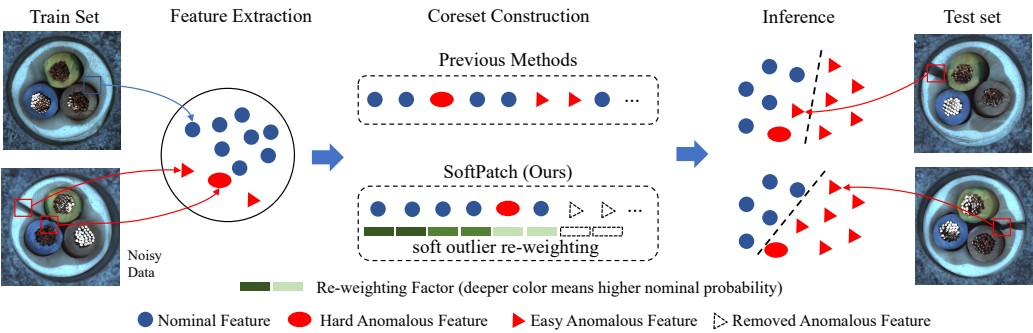

Figure 1: Illustration of SoftPatch. Unlike previous methods that construct coreset without considering the negative effect of noisy data, SoftPatch wipes off easy noisy data to formulate a clean training set and alleviates hard noisy data's impact by soft-reweighting.

it back at a random location. Similarity, DRAEM [15] synthesizes anomalies through Perlin Noise. Nevertheless, the inevitable discrepancy between the synthetic anomaly and the real anomaly disturbs the criteria of the model and limits the generalization performance. The gap between anomalies is usually larger than that between anomaly and normal. This is why AD methods deceived by some noisy samples can still work well when handling other kinds of anomalies.

**Agnostic methods.** Including knowledge distillation and image reconstruction, agnostic methods based on a theory that models that have never seen anomalies will behave differently in inference when inputting both normal and anomaly samples. Knowledge distillation is ingeniously used in anomaly detection. Bergmann et al. [16] propose that the representations of unusual patches are different between a pretrained teacher model and a student model, which tried its best to simulate teacher output with an anomaly-free training set. Based on this theory, Salehi et al. [17] propose that considering multiple intermediate outputs in distillation and using a smaller student network lead to a better result. Reverse distillation [18] uses a reverse flow that avoids the confusion caused by the same filters and prevents the propagation of anomaly perturbation to the student model, whose structure is similar to reconstruction networks. Image Reconstruction methods [7; 19; 20] utilize the assumption that the reconstruction network trained in the normal set can not reconstruct the anomaly part. A high-resolution result can be obtained by comparing the differences between the reconstructed and original images. However, all agnostic methods need long training stages, which limit their usage, i.e., the rapid deployment assumption in fully unsupervised learning.

**Feature modeling.** We specifically refer to the direct modeling of the output features of the extractor, including distribution estimation [21; 22], distribution transformation [23; 9], pre-trained model adaption [24; 25], and memory storage [26; 8]. PaDiM [21] utilizes multivariate Gaussian distributions to estimate the patch embedding of nominal data. In the inference stage, the embedding of irregular patches will be out of distribution. It is a simple but efficient method, but Gaussian distribution is inadequate for more complex data cases. So to enhance the estimation of density, DifferNet [23] and CFLOW [9] leverage the reversible normalizing flows based on multi-scale representation. Hou et al. [26] proposed that the granularity of division on feature maps is closely related to the reconstruction capability of the model for both normal and abnormal samples. So a multi-scale block-wise memory bank is embedded into an autoencoder network as a model of past data. PatchCore [8] is a more explicit but valuable memory-based method, which stores the sub-sampled patch features in the memory bank and calculates the nearest neighbor distance between the test feature and the coreset as an anomaly score. Although PatchCore is outperformance in the typical setting, it is overconfident in the training set, which leads to poor noise robustness.

## 2.2 Learning with Noisy Data

Noisy label recognition is becoming an emerging topic for supervised learning but has rarely been explored in unsupervised anomaly detection because there is no apparent label. For classification, some research [27; 28] propose to filter noisy pseudo-labeled data with a high confidence threshold. Li et al. [29] selects noisy-labeled data with a mixture model and trains in a semi-supervised manner.

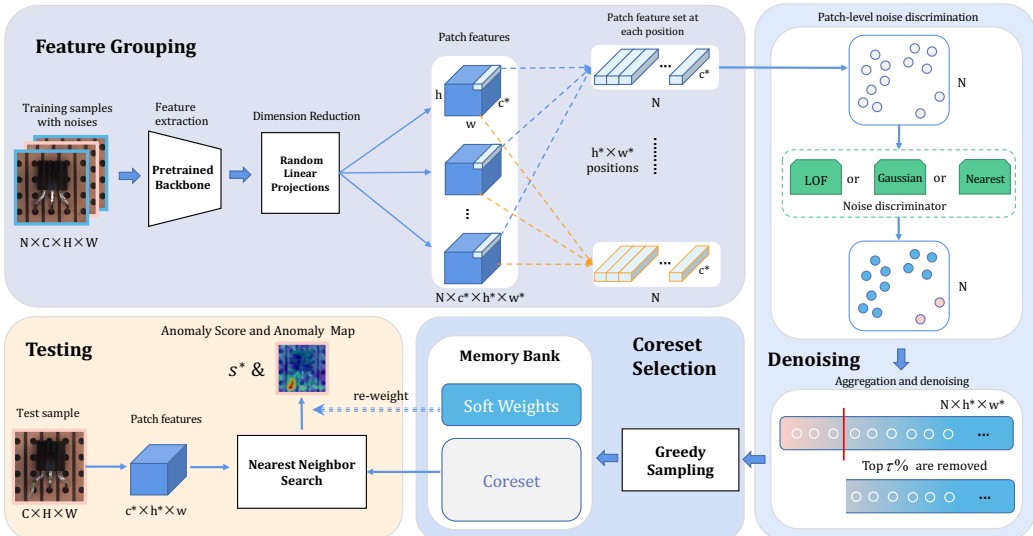

Figure 2: Overview of the proposed method. In the training phase, the noises are distinguished at patch level at each position of the feature map by a noise discriminator. The deeper color a patch node has, the higher probability that it is a noise patch. After achieving outlier scores for all patches, the top $\tau\%$ patches with the highest outlier score are removed. The coreset is a subset of remaining patches after denoising. Different from other methods, our memory bank consists of the samples in coreset and their outlier scores which are stored as soft weights. Soft weights will be further utilized to re-weight the anomaly score in inference.

Kong et al. [30] relabel harmful training samples. For object detection, multi-augmentation [31], teacher-student [32], or contrastive learning [33] are adopted to alleviate noise with the help of the expert model's knowledge. However, current noisy label recognition methods all rely on labeled data to co-rectify noisy data. In comparison, we target to improve the model's noise robustness in an unsupervised manner without introducing labor annotations.

While there are some model robustness researches on unsupervised AD, their objects and tasks are distinguished from our work since "anomaly detection" is an overloaded term. A recent survey [34] explores the model robustness of 30 AD algorithms. Nevertheless, unsupervised methods are excluded from the annotation errors setting. Pang et al. [35] deals with video anomaly without manually labeled data where information in consecutive frames can be exploited. While our work tackle anomaly detection from a single image. Other related papers [36; 37; 38] eliminate noisy and corrupted data in semantic anomaly detection. Unlike semantic anomaly detection, we focus on image sensory anomaly detection [1], which has recently raised much concern and contains a new task, anomaly localization. Although some existing methods [39; 40; 41; 42] treat covariate shift the same way they treat semantic shift and enhance model robustness with universal processes, their basic structures are poor compared with rapid-developed sensory AD methods, which leads to the robustness improvement insignificant. Noise in image sensory anomaly detection is more similar to the normal data and brings more challenges.

## 3 The Proposed Method

### 3.1 Overview

Patch-based unsupervised anomaly methods, such as PatchCore [8] and CFA [25], have three main processes: feature extraction, coreset selection with memory bank construction, and anomaly detection. One of the important assumptions is that the training set only contains nominal images, and the coreset should have full coverage of the entire training data distribution. During the test, an incoming image will directly search in the memory bank for similar features, and the anomaly score is the dissimilarity with the nearest patches. The searching process may collapse if the assumed

clean full coverage memory bank contains noise. Therefore, we propose SoftPatch, which filters noisy data by a noise discriminator before coreset construction and softens the searching process for down-weighting the hard unfiltered noisy samples.

The general denoising methods against the label contamination at the sample level are sub-optimal in image sensory anomaly detection. The abnormalities in image sensory AD, represented by industrial defect detection and medical image analysis, usually occupy only a tiny area of the image. At the sample level, noisy data is hard to distinguish, but the sample's inherent deviation may be more remarkable. So we propose a patch-level denoising strategy that works on the feature space to judge the noisy patch better. First, the collected features are grouped according to position to narrow the domain of noise discrimination. Then we insert the patch-level noise discrimination process before the coreset sampling, which generates the noise score according to the feature distribution of each position. Since most areas of the noisy image are usually anomaly-free, we remove those noisy patches and retain the rest to maximize the use of data. At the same time, the rest denoising scores reflecting the behavior of clustering are used to scale the anomaly score in inference. The other parts of the algorithm, such as feature extraction, dimension reduction, coreset sampling, and nearest neighbor search, follow the baseline PatchCore[8]. Figure 2 shows the framework of SoftPatch.

The target of image-level denoising is to find $\mathcal{X}_{noise}$ from $\mathcal{X}$, where $\mathcal{X} = \{x_i : i \in [1, N], x_i \in \mathbb{R}^{C \times H \times W}\}$ denotes training images (channels $C$, height $H$, width $W$). Following convention in existing work[8], we use $\phi_i \in \mathbb{R}^{c^* \times h^* \times w^*}$ as the feature map (channels $c^*$, height $h^*$, width $w^*$) of image $x_i \in \mathcal{X}$, $\phi_i(h, w) \in \mathbb{R}^{c^*}$ as the patch at $(h, w)$ on the aggregated feature map with dimension $c$.

### 3.2 Noise Discriminative Coreset Selection

With increasing training images, the features memory can become exceedingly large and infeasible to discriminate noise by overall statistics. Therefore, we group all features by position and count their outlier scores. Then all the scores are aggregated to determine noise patches, after which we just remove the features with top $\tau$ percent scores. We apply three noise reduction methods in total.

#### 3.2.1 Nearest Neighbor

With the assumption that the amount of noisy samples $X_{noise}$ is much less than clean samples $X_{nominal}$, we set Nearest neighbor distance as our baseline [43] where a large distance means an outlier. Given a set of images, $\phi \in \mathbb{R}^{N \times c^* \times h^* \times w^*}$ represents all features. Each patch's nearest neighbor distance $\mathcal{W}_i^{nn}$ is defined as:

$$\mathcal{W}_i^{nn}(h, w) = \min_{n \in [1,N], n \neq i} \|\phi_i(h, w) - \phi_n(h, w)\|_2, \tag{1}$$

We first calculate the distances, then take the minimum among batch dimensions (neighbor) as $W^{nn}$. Previous methods [8; 25] have proved that the minimum feature distance from a pretrained network can be an indicator to discriminate anomaly. This method can discriminate apparent outliers but suffer from uneven distribution of different clusters, where some clusters can have large inter-distance and lead to being mistakenly threshed as noisy data. To treat all clusters equally, we propose another multi-variate Gaussian method to calculate the outlier score without the interference of different clusters' densities.

#### 3.2.2 Multi-Variate Gaussian

With Gaussian's normalizing effect, all clean images' features can be treated equally. To apply Gaussian distribution on image characteristics dynamically, we calculate the inlier probabilities on the batch dimension for each patch $\phi_i(h, w)$, similar to 3.2.1. The multi-variate Gaussian distribution $\mathbb{N}(\mu_{h,w}, \Sigma_{h,w})$ can be formulated that $\mu_{h,w}$ is the batch mean of $\phi_i(h, w)$ and sample covariance $\Sigma_{h,w}$ is:

$$\Sigma_{h,w} = \frac{1}{N-1} \sum_{n=1}^{N} (\phi_n(h, w) - \mu_{h,w})(\phi_n(h, w) - \mu_{h,w})^T) + \epsilon I, \tag{2}$$

where the regularization term $\epsilon I$ makes $\sum_{h,w}$ full rank and invertible [21]. Finally, with the estimated multi-variate Gaussian distribution $\mathcal{N}(\mu_{h,w}, \sum_{h,w})$. Notice that we do not exclude the $x_i$ when

calculating the $\mathcal{N}$, so all samples can share a joint distribution, which is much less computationally intensive than doing the calculations one by one. Mahalanobis distance is calculated as the noisy magnitude $\mathcal{W}_i^{mvg}(h, w)$ of each patch:

$$\mathcal{W}_i^{mvg}(h, w) = \sqrt{(\phi_i(h, w) - \mu_{(h,w)})^T \Sigma_{h,w}^{-1}(\phi_i(h, w) - \mu_{(h,w)})}. \tag{3}$$

A high Mahalanobis distance means a high outlier score. Even though Gaussian distribution normalizes and captures the essence of image characteristics, small feature clusters may be overwhelmed by large feature clusters. In the scenario of a prominent feature cluster and a small cluster in a batch, the small cluster may be out of 1-, 2- or 3-$\Sigma$ of calculated $\mathbb{N}(\mu_{h,w}, \Sigma_{h,w})$ and erroneously classified as outliers. After analyzing the above two methods, we need a method that can: 1. treat all image characteristics equally; 2. treat large and small clusters equally; 3. high dimension calculation applicable.

### 3.2.3 Local Outlier Factor (LOF)

LOF[44] is a local-density-based outlier detector used mainly on E-commerce for criminal activity detection. Inspired by LOF, we can solve above mentioned three questions in 3.2.2: 1. Calculating the relative density of each cluster can normalize different density clusters; 2. Using local k-distance as a metric to alleviate the overwhelming effect of large clusters; 3. Modeling distance as normalized feature distance can be used on high-dimensional patch features. Therefore, the k-distance-based absolute local reachability density $lrd_i(h, w)$ is first calculated as:

$$lrd_i(h, w) = 1 / \left( \frac{\sum_{b \in \mathcal{N}_k(\phi_i(h,w))} dist_k^{reach}(\phi_i(h, w), \phi_b(h, w))}{|\mathcal{N}_k(\phi_i(h, w))|} \right), \tag{4}$$

$$dist_k^{reach}(\phi_i(h, w), \phi_b(h, w)) = max(dist_k(\phi_b(h, w)), d(\phi_i(h, w), \phi_b(h, w))), \tag{5}$$

where $d(\phi_i(h, w), \phi_b(h, w))$ is L2-norm, $dist_k(\phi_i(h, w))$ is the distance of kth-neighbor, $\mathcal{N}_k(\phi_i(h, w))$ is the set of k-nearest neighbors of $\phi_i(h, w)$ and $|\mathcal{N}_k(\phi_i(h, w))|$ is the number of the set which usually equal k when without repeated neighbors. With the local reachability density of each patch, the overwhelming effect of large clusters is largely reduced. To normalize local density to relative density for treating all clusters equally, the relative density $\mathcal{W}_i^{LOF}$ of image $i$ is defined below:

$$\mathcal{W}_i^{LOF}(h, w) = \frac{\sum_{b \in \mathcal{N}_k(\phi_i(h,w))} lrd_b((h, w))}{|\mathcal{N}_k(\phi_i(h, w))| \cdot lrd_i(h, w)}. \tag{6}$$

$\mathcal{W}_i^{LOF}(h, w)$ is the relative density of the neighbors over patch's own, and represents as a patch's confidence of inlier. Our experiments found that all three noise reduction methods above are helpful in data pre-selection before coreset construction, while $LOF$ provides the best performance. However, after visualization of our cleaned training set, we found that hard noisy samples, which are similar to nominal samples, are still hidden in the dataset. To further alleviate the effect of noisy data, we propose a soft re-weighting method that can down-weight noisy samples according to outlier scores.

### 3.3 Anomaly Detection based on SoftPatch

Besides the construction of the Coreset, outlier factors of all the selected patches are stored as soft weights in the memory bank. With the denoised patch-level memory bank $\mathcal{M}$ as shown in figure 2, the image-level anomaly score $s \in \mathbb{R}$ can be calculated for a test sample $x_i \in \mathcal{X}^{test}$ by nearest neighbor searching at patch level. Denoting the collection of patch features of a test sample as $\mathcal{P}(x_i)$, for each patch $p_{h,w} \in \mathcal{P}_{x_i}$ the nearest neighbor searching can be formulated as the following equation:

$$m^* = \underset{m \in \mathcal{M}}{\arg\min} \|p - m\|_2. \tag{7}$$

After nearest searching, pairs of test patch and its corresponding nearest neighbor in $\mathcal{M}$ can be achieved as $(p, m^*)$. For each patch $p_{i,j} \in \mathcal{P}_{x_i}$, the patch-level anomaly score is calculated by

$$s_{h,w} = \mathcal{W}_{m^*} \|p_{h,w} - m^*\|_2, \tag{8}$$

where $\mathcal{W}_{m^*}$ is the soft weight calculated by one noise discriminator. The image-level anomaly score is attained by finding the largest soft weights re-weighted patch-level anomaly score: $s^* = \underset{(h,w)}{\max} s_{h,w}$.

Table 1: Anomaly detection performance on MVTecAD with noise. The results are evaluated on MVTecAD-noise-0.1. *Overlap* means the injected anomalous images are included in the test set. PaDiM* uses ResNet18 as the backbone. PatchCore-random uses 1% random subsampler instead of the default greedy subsampler. *Gap* row shows the performance gap between a noisy scene and a normal scene.

| Noise=0.1 | | | No overlap | | | | | | Overlap | |
|---|---|---|---|---|---|---|---|---|---|---|
| Category | PaDiM | CFLOW | PatchCore | SoftPatch-nearest | SoftPatch-gaussian | SoftPatch-lof | PaDiM* | PatchCore | PatchCore-random | SoftPatch-lof |
| bottle | 0.994 | 0.998 | **1.000** | **1.000** | 0.997 | 0.937 | **1.000** | 0.692 | 0.998 | **1.000** |
| cable | 0.873 | 0.925 | 0.982 | 0.935 | 0.952 | **0.995** | 0.680 | 0.756 | 0.920 | **0.994** |
| capsule | 0.920 | 0.947 | **0.976** | 0.916 | 0.662 | 0.963 | 0.796 | 0.783 | 0.779 | **0.955** |
| carpet | **0.999** | 0.961 | 0.996 | 0.995 | **0.999** | 0.991 | 0.890 | 0.681 | 0.973 | **0.993** |
| grid | 0.966 | 0.891 | 0.971 | 0.972 | **0.997** | 0.968 | 0.674 | 0.526 | 0.793 | **0.969** |
| hazelnut | 0.956 | **1.000** | 0.998 | **1.000** | **1.000** | **1.000** | 0.543 | 0.441 | 0.998 | **1.000** |
| leather | **1.000** | **1.000** | **1.000** | **1.000** | **1.000** | **1.000** | 0.964 | 0.739 | **1.000** | **1.000** |
| metal_nut | 0.987 | 0.959 | **0.999** | 0.994 | 0.997 | **0.999** | 0.820 | 0.765 | 0.969 | **1.000** |
| pill | 0.918 | 0.929 | **0.975** | 0.921 | 0.873 | 0.963 | 0.722 | 0.770 | 0.874 | **0.955** |
| screw | 0.838 | 0.784 | **0.966** | 0.862 | 0.475 | 0.960 | 0.567 | 0.710 | 0.462 | **0.923** |
| tile | 0.977 | 0.991 | 0.985 | 0.996 | **0.997** | 0.993 | 0.830 | 0.716 | **1.000** | 0.981 |
| toothbrush | 0.927 | 0.906 | 0.997 | **1.000** | 0.997 | 0.997 | 0.700 | 0.800 | 0.797 | **0.994** |
| transistor | 0.953 | 0.896 | 0.953 | **1.000** | 0.992 | 0.990 | 0.471 | 0.491 | 0.943 | 0.999 |
| wood | 0.991 | 0.972 | 0.984 | 0.984 | **0.997** | 0.987 | 0.831 | 0.579 | 0.980 | **0.986** |
| zipper | 0.852 | 0.928 | **0.981** | 0.976 | 0.979 | 0.978 | 0.679 | 0.792 | 0.950 | **0.974** |
| Average | 0.943 | 0.939 | 0.984 | 0.970 | 0.927 | **0.986** | 0.740 | 0.683 | 0.896 | **0.982** |
| Gap | -0.007 | -0.03 | -0.008 | **+0.002** | -0.001 | 0.0 | -0.151 | -0.309 | -0.015 | **-0.004** |

Different from PatchCore which directly considers patches equally, SoftPatch softens anomaly scores by noise level from noise discriminater. The soft weights, i.e., local outlier factors, have considered the local relationship around the nearest node. Thus, a similar effect can be achieved as PatchCore but with more noise robustness and fewer searches. According to the image-level anomaly score, a sample is classified into a normal sample or an abnormal sample.

## 4 Experiments

### 4.1 Experimental Details

**Datasets.** Our experiments are mainly conducted on the MVTecAD and BTAD benchmarks[11; 12]. MVTecAD contains 15 categories with 3629 training images and 1725 test images in total, and BTAD has three categories with 1799 images, where different classes of industry production mean a comprehensive challenge, such as object or texture and whether rotation. Since each category of MVTecAD is divided into nominal-only images and a test set with both nominal and anomalous samples, to create a noisy training set, we sample anomalous images randomly from the test set and mix them with the existing training images. Notice that the original normal number of samples in the training set remains unchanged compared with the noiseless case. In this setting(*No overlap*), the injected anomalous samples will not be evaluated, which is more likely the case in the real application. We also construct a different setting(*Overlap*) where the injected anomalous samples are also in the test set to demonstrate the risk that defects with similar appearance will severely exacerbate the performance of an anomaly detector trained with noisy data. Meanwhile, the overlap samples test the outlier detection performance of our algorithm. By controlling the proportion of negative samples being injected into the train set, we obtain several new datasets with different noise ratios dubbed MVTecAD-noise-$n$, where $n$ refers to the ratio of noise. For BTAD, we just use the original fold.

**Evaluation Metrics.** We report both image-level and pixel-level AUROC for each category in MVTecAD and average them to get the average image/pixel level AUROC. In order to represent noise robustness, the performance gaps between noise-free data and noisy data are also displayed. When not otherwise stated, our method SoftPatch refers to SoftPatch-LOF that uses LOF in Section 3.2.3.

**Implementation Details.** We test three SOTA AD algorithms, PatchCore [8], PaDim [21] and CFLOW [9] in noise scene and follow their main settings. In the absence of specific instructions, the backbone of the feature extractor is *Wide-ResNet50*, and the coreset sampling ratio of PatchCore and SoftPatch is 10%. For MVTecAD images, we only use $256 \times 256$ resolution and center crops

Table 2: Anomaly localization performance on MVTecAD with noise. The results are evaluated on MVTecAD-noise-0.1.

| Noise=0.1 | | No overlap | | | | | | | Overlap | | |
|---|---|---|---|---|---|---|---|---|---|---|---|
| Category | PaDiM | CFLOW | PatchCore | SoftPatch-nearest | SoftPatch-gaussian | SoftPatch-lof | PaDiM* | PatchCore | PatchCore-random | SoftPatch-lof |
| Average | 0.972 | 0.969 | 0.956 | 0.971 | 0.977 | **0.979** | 0.955 | 0.654 | 0.951 | **0.969** |
| Gap | -0.007 | -0.006 | -0.025 | -0.008 | **-0.001** | -0.002 | -0.013 | -0.327 | -0.021 | **-0.012** |

Table 3: Anomaly detection performance on BTAD without additional noise. The best results are in bold, and the second-best results are underlined. The last column lists the count of anomaly samples in the test set.

| Category | SPADE | P-SVDD | PatchCore | PaDiM | SoftPatch(ours) | Anomaly samples |
|---|---|---|---|---|---|---|
| 01 | 0.914 | 0.957 | **1.000** | **1.000** | 0.999 | 50 |
| 02 | 0.714 | 0.721 | 0.871 | 0.871 | **0.934** | 200 |
| 03 | **0.999** | 0.821 | **0.999** | 0.971 | 0.997 | 41 |
| Mean | 0.876 | 0.833 | 0.957 | 0.947 | **0.977** | - |

them into $224 \times 224$ along with a normalization. For BTAD, we use $512 \times 512$ resolution. We train a separate model for each class. Notice that unlike many methods setting the hyperparameters according to the noise ratio, which is unknowable in reality, we set the threshold $\tau$ in SoftPatch and the *LOF-K* to constant 0.15 and 6 for all noisy scenarios and classes. The effects of hyperparameters are studied in the ablation study. All our experiments are run on Nvidia V100 GPU and repeated three times to report the average results.

## 4.2 Anomaly Detection Performance with Noise

**Experiments on MVTecAD.** As indicated in Table 1 and Table 2, when 10% of anomalous samples are added to corrupt the train set, all existing methods have different extents of performance decrease, although not disastrously in *No overlap* setting. Compared to other methods, the proposed SoftPatch exhibits much stronger robustness against noisy data both in terms of anomaly detection and localization, no matter which noise discriminator is used. Among three variants of SoftPatch, SoftPatch-lof achieves the best overall performance with the highest accuracy and strongest robustness. Interestingly, PaDiM[21], CFLOW[9] and SoftPatch-gaussian show significantly less performance drop than PatchCore, which indicates that modeling feature as Gaussian distribution does help denoising. While modeling feature distribution at each spatial location as a single Gaussian distribution can't handle misaligned images, such as *screw* class in MVTecAD, which explains the poor performance on these classes(see screw row). On the other hand, PatchCore's greedy-sampling strategy is a double-edged sword with higher feature space coverage and higher sensitivity to noise. That's why using random sampling in PatchCore is more robust with compromised performance(see PatchCore 1%-Random column). SoftPatch-nearest does a slightly better job in the misaligned cases. However, it doesn't take feature distribution into account, which leads to inferior performance.

**Experiments on BTAD.** We also compare SoftPatch with other SOTA methods on another dataset, BTAD. Surprisingly, SoftPatch gives out a new SOTA result, even in the original setting that contains no additional noise (Table 3). By reviewing all the training samples, we find that there are already many noisy samples (usually small scratches) in the training set of category BTAD-02, which is more consistent with our setting and further demonstrates the necessity of our approach. The noisy images are provided in Appendix A.6 (Table 8). Moreover, the BTAD-02 contains more anomaly samples with similar appearance anomalies. In the category of BTAD-02, our method attains significant improvement compared to others. SoftPatch can also maintain the leading performance if the noise is added artificially(Appendix A.8).

**Performance trends.** In order to explore how different methods behave with the increasing noise level, experiments are further performed on MVTecAD-noise-$\{0 \sim 0.15\}$. The results of the proposed methods are provided in Figure 3. Under the *No overlap* setting, as the noise ratio increases, PatchCore shows a pixel-level AUROC drop up to 3.7%. The performance decreases as the noise ratio rises. On the contrary, although the default performance is slightly poorer than PatchCore(about 0.006

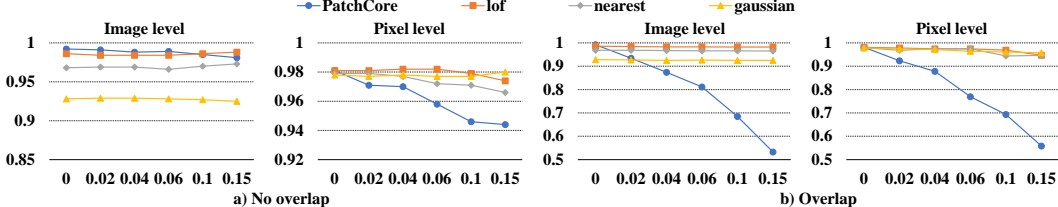

Figure 3: The comparison of anomaly detection performance under noisy training. *no overlap* means the injected anomalous images are removed from test set while *overlap* are not.

and 0 decrease in image-level and pixel-level AUROC), the proposed SoftPatch-lof deteriorates much slower, which demonstrates better denoising ability. As for SoftPatch-nearest and SoftPatch-gaussian, they are also more robust, however, with worse base performance(see Figure 3 at noise ratio=0). The visualization of the coreset in Figure 5 also shows that random sampling avoids sampling the outlier but can not model normal adequately. Being consistent with the discussion above, under the *Overlap* setting, PatchCore's performance is getting worse and worse catastrophically(up to 40% AUROC drop in both image and pixel level) as more noises are added. This is expectable since PatchCore uses a greedy strategy for coreset sampling, which favors outliers in feature space. SoftPatch-lof consistently outperforms other methods with no significant performance drop as the noise level goes up. Appendix A.3 (Figure 6 and 7) shows more comparison with others. The experimental results indicate that the risk is hidden by the fact that defects in MVTecAD have very different appearances. In this case, even if some anomalous features are added mistakenly to the coreset, they are unlikely to be retrieved during test time. However, the risk still exists and will be triggered when similar defects show up at test time.

More experiments can be found in appendix, such as the comparison of image-level and patch-level denoising(Appendix A.4), computational analysis(Appendix A.5) and an augmented overlap setting(Appendix A.7).

Table 4: The ablation study of soft weight. The performance scores are *Image/pixel-level AUROC* on MVTecAD.

| Noise discriminator | Soft weight | No overlap | | Overlap | |
|---|---|---|---|---|---|
| | | Image level | Pixel level | Image level | Pixel level |
| None | | 0.985 | 0.946 | 0.685 | 0.693 |
| Gaussian | | 0.927 | 0.977 | 0.925 | 0.961 |
| Gaussian | ✓ | 0.922 | 0.974 | 0.924 | 0.965 |
| Nearest | | 0.970 | 0.971 | 0.966 | 0.944 |
| Nearest | ✓ | 0.972 | 0.978 | 0.968 | 0.958 |
| LOF | | 0.985 | **0.984** | **0.984** | 0.963 |
| LOF | ✓ | **0.986** | 0.979 | 0.982 | **0.969** |

## 4.3 Ablation Study

### 4.3.1 Effectiveness of the Proposed Modules

We validated the effectiveness of two proposed modules **noise discriminator** and **soft weight** by removing them from the pipeline. As shown in Table 4, the noise discriminator significantly improves the noise robustness in terms of pixel-level AUROC. Among three decision choices of noise discriminator, LOF achieved the best balance between robustness and capacity, resulting in the most performance boost under all settings. We further analyzed the intermediate results by visualizing the sampled coreset of different methods, which shows that SoftPatch-LOF sampled much fewer anomalous features than the baseline(see Figure 5). Soft weight is used alongside the noise discriminator to further improve the final results. We only observed minor improvement for using Soft weight in SoftPatch-Nearest. We suspect that the other two kinds of noise discriminators are already robust against noise data.

Table 5: *Image/pixel-level AUROC result for different LOF-K on two settings.*

| K | 3 | 4 | 5 | 6 | 7 | 8 | 9 |
|---|---|---|---|---|---|---|---|
| Overlap | **0.983**/0.955 | 0.982/0.951 | **0.983**/0.959 | 0.982/**0.975** | 0.981/0.973 | 0.982/0.968 | 0.980/0.968 |
| No overlap | **0.985**/0.972 | **0.985**/0.975 | 0.984/0.977 | 0.984/0.982 | **0.985**/0.980 | 0.984/**0.983** | 0.981/0.982 |

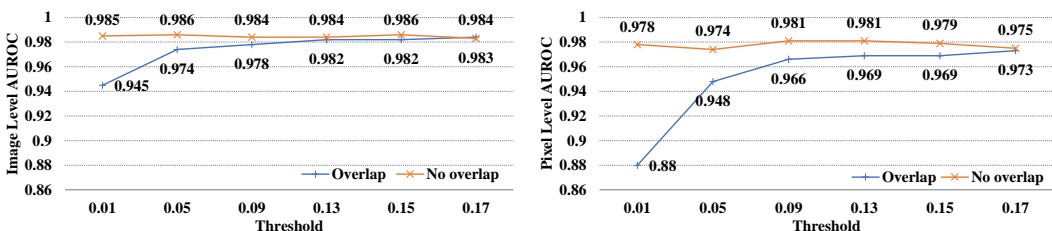

Figure 4: Performance trend with the threshold $\tau$ in SoftPatch-LOF. The results are evaluated on MVTecAD-noise-0.1.

### 4.3.2 Parameter Selection

To explore the impact of two parameters (*LOF-k* and threshold $\tau$) on the final performance, we perform parameters searching on our method. As in Table 5, our method achieves better performance when *LOF-k* is greater than 5, which suggests that our method is not sensitive to *LOF-k*, as long as it is not too small or too large. If *LOF-k* is too small, it fails to estimate the local density accurately because too few neighbors are considered. On the contrary, a large *LOF-k* may lead to undesirable cross-clusters connection that can not capture real data distribution.

Threshold $\tau$ refers to the ratio of eliminated patch features when building coreset. Figure 4 indicates an increasing trend of AUROC as threshold $\tau$ increases under *Overlap* setting, which is expected since a higher threshold means a more aggressive denoising strategy. In *Overlap* setting, the mistakenly sampled features are the direct reason for the drastic performance drop. Therefore more aggressive denoising improves the result significantly. However, In *No Overlap* setting, the effect of the noisy feature is less prominent. Although the best *LOF-k* and threshold $\tau$ are changed according to the class and noise level, we simply use fixed values, 6 and 0.15, in all situations.

## 5 Conclusions

This paper emphasizes the practical value of investigating noisy data problems in unsupervised AD. Introducing a novel noisy setting on the previous task, we test the performance of existing methods and SoftPatch. For existing methods, despite no adaptation to noisy settings, some of them have a slight performance decrease in some scenes. However, the performance decrease could be more significant and catastrophic for other methods or in other scenes. For the proposed SoftPatch, it shows consistent performance in all noise settings, which outperforms other methods.

Industrial inspection systems are an important computer vision application that requires good robustness. The noise injected into the training set break with the naive assumption that the training samples were normal. Noise also gives the model early exposure to the distribution of anomalies. The unsupervised AD with noisy data needs more research in the future.

## Acknowledgement

This work is supported by the National Natural Science Foundation of China under Grant No. 61972188, 62122035 and 62206122.

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
