# A Appendix

## A.1 Visualization of Coreset

Figure 5 shows the visualization of coreset in the memory bank. PatchCore reserve too many noisy features, which are obviously outliers. Though replacing the greedy sampling with random sampling, PatchCore avoids most noisy features but is poor at model training set and still misled by some noise. The coreset of SoftPatch is clean and decentralized. Our coreset saves some features from noisy samples because we believe that abnormal images also contain a large number of normal patches. So the features conforming to the normal distribution are reserved to enhance the model perception.

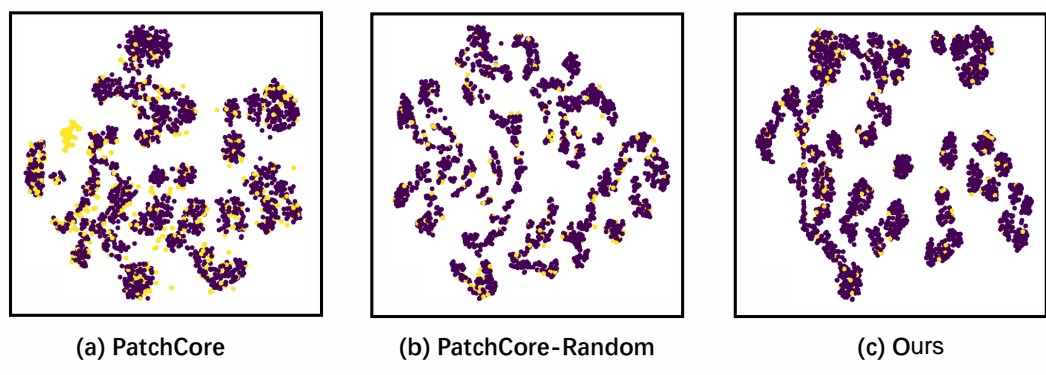

|       (a) PatchCore       |       (b) PatchCore-Random       |       (c) Ours       |

Figure 5: Comparison between corsets of AD methods with same noisy train set, MVTecAD-Pill with noise-0.1. We use t-SNE for dimension reduction for visualization. The yellow dots represent patch features from noisy sample, while the purple dots are nominal. Compared with the other two, SoftPatch wipe off the noisy patch and model the nominal data properly.

## A.2 Details of Experimental Results

Table 6: Anomaly localization performance details of all classes. The results are evaluated on MVTecAD-noise-0.1.

| Noise=0.1 | No overlap | | | | | | Overlap | | | |
|---|---|---|---|---|---|---|---|---|---|---|
| Category | PaDiM | CFLOW | PatchCore | SoftPatch-nearest | SoftPatch-gaussian | SoftPatch-lof | PaDiM* | PatchCore | PatchCore-random | SoftPatch-lof |
| bottle | 0.986 | 0.984 | 0.987 | 0.987 | 0.986 | 0.987 | 0.981 | 0.714 | 0.979 | 0.975 |
| cable | 0.916 | 0.958 | 0.843 | 0.915 | 0.981 | 0.983 | 0.946 | 0.670 | 0.969 | 0.971 |
| capsule | 0.986 | 0.985 | 0.986 | 0.988 | 0.977 | 0.990 | 0.984 | 0.883 | 0.984 | 0.989 |
| carpet | 0.992 | 0.989 | 0.992 | 0.992 | 0.993 | 0.992 | 0.980 | 0.765 | 0.951 | 0.989 |
| grid | 0.974 | 0.947 | 0.991 | 0.990 | 0.989 | 0.990 | 0.879 | 0.482 | 0.882 | 0.974 |
| hazelnut | 0.987 | 0.991 | 0.990 | 0.990 | 0.991 | 0.990 | 0.978 | 0.418 | 0.957 | 0.924 |
| leather | 0.994 | 0.994 | 0.991 | 0.994 | 0.994 | 0.993 | 0.992 | 0.683 | 0.987 | 0.993 |
| metal_nut | 0.933 | 0.956 | 0.842 | 0.894 | 0.964 | 0.984 | 0.911 | 0.779 | 0.938 | 0.983 |
| pill | 0.956 | 0.983 | 0.971 | 0.974 | 0.972 | 0.981 | 0.960 | 0.608 | 0.971 | 0.976 |
| screw | 0.989 | 0.977 | 0.995 | 0.991 | 0.969 | 0.994 | 0.974 | 0.745 | 0.953 | 0.969 |
| tile | 0.956 | 0.953 | 0.953 | 0.960 | 0.962 | 0.954 | 0.921 | 0.700 | 0.919 | 0.954 |
| toothbrush | 0.991 | 0.988 | 0.989 | 0.988 | 0.988 | 0.985 | 0.954 | 0.692 | 0.984 | 0.985 |
| transistor | 0.960 | 0.887 | 0.847 | 0.965 | 0.954 | 0.942 | 0.939 | 0.317 | 0.914 | 0.936 |
| wood | 0.973 | 0.964 | 0.969 | 0.947 | 0.946 | 0.939 | 0.946 | 0.522 | 0.896 | 0.929 |
| zipper | 0.986 | 0.978 | 0.986 | 0.989 | 0.988 | 0.988 | 0.978 | 0.823 | 0.975 | 0.986 |
| Average | 0.972 | 0.969 | 0.956 | 0.971 | 0.977 | **0.979** | 0.955 | 0.654 | 0.951 | **0.969** |
| Gap | -0.007 | -0.006 | -0.025 | -0.008 | **-0.001** | -0.002 | -0.013 | -0.327 | -0.021 | **-0.012** |

## A.3 Performance Trends in Noise

Figure 6 and 7 show the performance trends of SOTA AD methods and SoftPatch in different noisy scenes. Since overconfident in the training data and the greedy subsampling algorithm, PatchCore performance decreases most obviously with the noise increase. In contrast, CFLOW and PaDiM

are also affected by noise, but the amplitudes are smaller. SoftPatch maintains a consistent level of performance at all noise levels. Unfortunately, SoftPatch is slightly weaker than PatchCore in noiseless scenes, which may be due to the excessively conservative threshold setting.

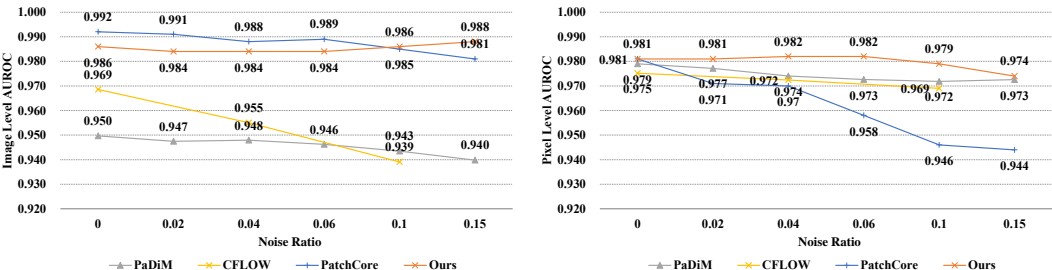

Figure 6: Performance in different level of no overlap noise.

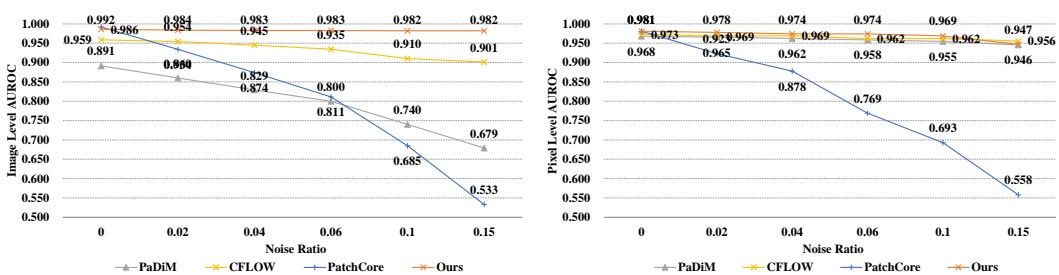

Figure 7: Performance in different level of overlap noise.

## A.4   Image-level Denoising V.S. Patch-level Denoising

A simple strategy to eliminate the noisy data is to delete the anomaly samples before training, which is an unsupervised outlier detection task. However, the existing outlier detection methods do not work well because the distance between abnormal and normal images is much smaller than the distance between different classes. Meanwhile, we found that some AD methods could also detect outliers in the training set. Following [41], we apply PaDiM* as the image-level denoising method which give consideration to the costs and effects. PaDiM* is a simplified version of PaDiM, which uses ResNet18 as the backbone with faster computing speed. PaDiM* scores all training samples and then removes the pieces with high outliers based on the threshold. The comparison in Table 7 and Table 8 show that image-level denoising dramatically improves the performance of existing SOTA AD methods in the noisy scene. But there is still a gap when compared with SoftPatch.

## A.5   Computational Analysis

SoftPatch does not require more runtime than PatchCore, according to theoretical analysis. The complexity of the greedy sampling process in PatchCore is $\mathcal{O}(N^2 h^2 w^2)$, which is most expensive part. The complexity of the noise discrimination process in SoftPatch-LOF is $\mathcal{O}(N^2 hw)$, since features are grouped before. So the computational complexity of SoftPatch is equal PatchCore by $\mathcal{O}(N^2 hw + N^2 h^2 w^2) = \mathcal{O}(N^2 h^2 w^2)$. In fact, SoftPatch will be faster because it removes a part of the patch as noise.

Excluding the loading time of data, the comparison of the remaining time overhead between SoftPatch and PatchCore is shown in Figure 9. The GPU used in this experiment is RTX TITAN 24G. Both spend almost the same amount of time training and testing, which means that our patch-level denoising does not bring unacceptable overhead. On the contrary, the image-level denoising dramatically increases training time.

Table 7: The anomaly detection performance of image-level denoising and patch-level denoising. The PaDiM*+PaDiM*, PaDiM*+CFLOW, and PaDiM*+PatchCore are AD methods with image-level denoising. PaDiM* is used in image level denoising, where we use the same threshold (0.15) as it in SoftPatch. And we also tried the tricky threshold-0.1 as the noise ratio, but it works worse. The results are evaluated on MVTecAD-noise-0.1 with overlap.

| Category | PaDiM* | PaDiM*+ PaDiM* | CFLOW | PaDiM*+ CFLOW | PatchCore | PaDiM*(threshold -0.1)+PatchCore | PaDiM*+ PatchCore | SoftPatch-lof |
|---|---|---|---|---|---|---|---|---|
| bottle | 0.937 | 0.994 | **1.000** | **1.000** | 0.692 | 0.984 | **1.000** | **1.000** |
| cable | 0.680 | 0.741 | 0.916 | 0.841 | 0.756 | 0.890 | 0.888 | **0.994** |
| capsule | 0.796 | 0.854 | 0.945 | 0.939 | 0.783 | 0.892 | 0.909 | **0.955** |
| carpet | 0.890 | 0.937 | 0.960 | 0.950 | 0.681 | 0.963 | 0.974 | **0.993** |
| grid | 0.674 | 0.765 | 0.799 | 0.830 | 0.526 | 0.850 | 0.870 | **0.969** |
| hazelnut | 0.543 | 0.725 | 0.999 | 0.990 | 0.441 | 0.871 | 0.929 | **1.000** |
| leather | 0.964 | 0.979 | 0.996 | **1.000** | 0.739 | 0.957 | 0.989 | **1.000** |
| metal_nut | 0.820 | 0.949 | 0.957 | 0.986 | 0.765 | 0.965 | 0.977 | **1.000** |
| pill | 0.722 | 0.745 | 0.897 | 0.924 | 0.770 | 0.898 | 0.913 | **0.955** |
| screw | 0.567 | 0.542 | 0.570 | 0.639 | 0.710 | 0.916 | 0.907 | **0.923** |
| tile | 0.830 | 0.906 | 0.980 | 0.981 | 0.716 | 0.939 | 0.957 | **0.981** |
| toothbrush | 0.700 | 0.869 | 0.878 | 0.928 | 0.800 | 0.981 | **0.997** | 0.994 |
| transistor | 0.471 | 0.770 | 0.872 | 0.788 | 0.491 | 0.777 | 0.825 | **0.999** |
| wood | 0.831 | 0.966 | 0.954 | 0.970 | 0.579 | 0.943 | 0.976 | **0.986** |
| zipper | 0.679 | 0.678 | 0.931 | 0.873 | 0.792 | 0.909 | 0.914 | **0.974** |
| Average | 0.740 | 0.828 | 0.910 | 0.909 | 0.683 | 0.916 | 0.935 | **0.982** |

Table 8: The anomaly localization performance of image-level denoising and patch-level denoising.

| Category | PaDiM* | PaDiM*+ PaDiM* | CFLOW | PaDiM*+ CFLOW | PatchCore | PaDiM*(threshold -0.1)+PatchCore | PaDiM*+ PatchCore | SoftPatch-lof |
|---|---|---|---|---|---|---|---|---|
| bottle | 0.981 | 0.983 | 0.984 | **0.986** | 0.714 | 0.984 | 0.985 | 0.975 |
| cable | 0.946 | 0.954 | 0.950 | 0.956 | 0.670 | 0.738 | 0.739 | **0.971** |
| capsule | 0.984 | 0.982 | 0.986 | 0.985 | 0.883 | 0.851 | 0.876 | **0.989** |
| carpet | 0.980 | 0.984 | 0.986 | 0.988 | 0.765 | 0.960 | 0.988 | **0.989** |
| grid | 0.879 | 0.876 | 0.961 | 0.948 | 0.482 | 0.797 | 0.818 | **0.974** |
| hazelnut | 0.978 | 0.977 | 0.982 | **0.987** | 0.418 | 0.798 | 0.825 | 0.924 |
| leather | 0.992 | 0.993 | 0.993 | **0.995** | 0.683 | 0.966 | 0.979 | 0.993 |
| metal_nut | 0.911 | 0.968 | 0.960 | **0.984** | 0.779 | 0.784 | 0.834 | 0.983 |
| pill | 0.960 | 0.956 | 0.976 | **0.984** | 0.608 | 0.706 | 0.713 | 0.976 |
| screw | **0.974** | 0.968 | 0.973 | 0.970 | 0.745 | 0.887 | 0.889 | 0.969 |
| tile | 0.921 | 0.927 | 0.945 | 0.946 | 0.700 | 0.924 | **0.968** | 0.954 |
| toothbrush | 0.954 | **0.986** | 0.984 | 0.983 | 0.692 | 0.977 | 0.986 | 0.985 |
| transistor | 0.939 | **0.965** | 0.834 | 0.908 | 0.317 | 0.932 | 0.945 | 0.936 |
| wood | 0.946 | **0.947** | 0.934 | 0.943 | 0.522 | 0.800 | 0.918 | 0.929 |
| zipper | 0.978 | 0.973 | 0.979 | 0.967 | 0.823 | 0.875 | 0.878 | **0.986** |
| Average | 0.955 | 0.963 | 0.962 | **0.969** | 0.654 | 0.865 | 0.889 | **0.969** |

Table 9: Mean training and inference time per category on MVTecAD. The unit of time is second.

| | Training time | Inference time |
|---|---|---|
| SoftPatch-LOF | 21.2958 | 15.6146 |
| PatchCore | 21.3869 | 15.8763 |
| PaDiM*+PatchCore | 74.2912 | 15.5386 |

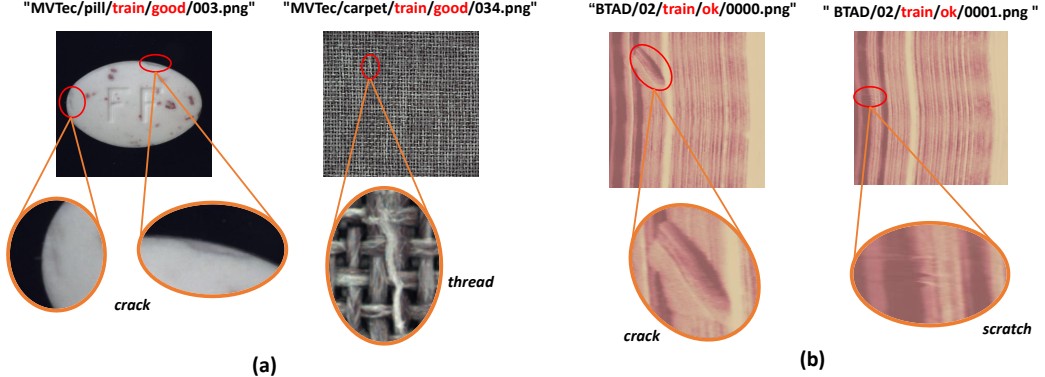

Figure 8: Noisy examples in (a) MVTecAD dataset and (b) BTAD dataset.

Table 10: Performance on MVTecAD in augmented *overlap* setting.

| Setting | Overlap with gaussian noise | Overlap with noise and blur | Overlap with rotation | Overlap with affine transformation |
|---|---|---|---|---|
| Method | PatchCore / Ours | PatchCore / Ours | PatchCore / Ours | PatchCore / Ours |
| Detection | 0.760 / **0.984** | 0.848 / **0.984** | 0.950 / **0.984** | 0.933 / **0.984** |
| Localization | 0.790 / **0.969** | 0.864 / **0.970** | 0.924 / **0.978** | 0.915 / **0.978** |

## A.6 The Noise in Existing Datasets

Although existing research datasets are well organized, some abnormal samples are misclassified. Fig. 8 show anomaly samples in normal set in two wide-used datasets. In the actual production data, the noise interference will be more serious.

## A.7 Performance in Augmented Overlap Setting

We do another experiment where the overlap images are augmented in the train set to make them different from the images in the test set. We experiment with varying degrees of appearance and structural augmentation. The result in Table 10 shows that our method still presents better robustness when the overlap samples have been transformed, though the performance of PatchCore is improved.

## A.8 Performance on BTAD with noise

The performance comparisons are provided in table 11 and 12. Since the anomaly samples in category BTAD-03 are not enough to meet the requirement of the number of noise samples, we experience the other two.

Table 11: Anomaly detection performance on BTAD-noise-0.1.

| Noise = 0.1 | No overlap | | Overlap | |
|---|---|---|---|---|
| **Category** | PatchCore | SoftPatch-LOF | PatchCore | SoftPatch-LOF |
| 01 | **1.000** | **1.000** | 0.522 | **1.000** |
| 02 | 0.860 | **0.922** | 0.738 | **0.912** |
| Mean | 0.930 | **0.961** | 0.630 | **0.956** |

Table 12: Anomaly localization performance on BTAD-noise-0.1.

| Noise = 0.1 | No overlap | | Overlap | |
|---|---|---|---|---|
| **Category** | PatchCore | SoftPatch-LOF | PatchCore | SoftPatch-LOF |
| 01 | 0.982 | **0.999** | 0.319 | **0.815** |
| 02 | 0.949 | **0.953** | 0.754 | **0.936** |
| Mean | 0.966 | **0.976** | 0.536 | **0.875** |