# OpenReview forum: "SoftPatch: Unsupervised Anomaly Detection with Noisy Data"
_NeurIPS.cc/2022/Conference — NeurIPS 2022 Accept_

### Official Review · Reviewer_zKJo · 2022-07-11

**Rating:** 8
**Confidence:** 4
**Soundness:** 3 good
**Presentation:** 3 good
**Contribution:** 3 good

**Summary:**

This paper addresses a practical setting of Anomaly Detection, namely AD with noisy data. The authors propose a novel AD algorithm named SoftCore to enhance model performance and noise robustness. By using a patch-level selection strategy, the model can reweight the coreset samples and wipe off the noisy image patch of noisy samples.

**Questions:**

Please refer to the weakness

**Limitations:**

Yes.

**Strengths And Weaknesses:**

Strengths:
+ The paper is well-structured, and easy to follow.
+ This paper explores the impact of noise data on anomaly detection tasks, which has highly practical significance.
+ The proposed task setup is reasonable, and the authors have conducted fundamental experiments to explain insights and analyze the impact of noise on AD tasks.
+ From the experimental results, the effectiveness of the proposed method is verified.

Weaknesses:
- There are some grammatical errors: Line 156, $W_(i)$ should be $W_{(i)}$; Line 268, LOF achieve -> achieved.
- The mathematical notation should be improved. For example, the symbol of sample variance $\sum_{h,w}$ is confusing in Eq(2) and (3).
- The critical threshold τ should be given more explanations.

---

> ### Author Response · Authors · 2022-08-02
> **The Response to Reviewer zKJo**
>
> Thanks for your positive comments and insightful suggestions. Please find our response below.
> ### Q1: There are some grammatical errors.
> Thank you for pointing out our fault. We will fix these in the revised version.
> ### Q2: The mathematical notation should be improved. For example, the symbol of sample variance $\Sigma_{h, w}$ is confusing in Eq(2) and (3).
> Thank you for pointing out the confusing notation. We will check all formulas and change the notation of sample covariance as follows.
>
> ${\Sigma_{h, w}}= \frac{1}{b^* - 1}\sum_{b=1}^{b^*}(\phi_b(h, w) - \mu_{h, w})(\phi_b(h, w) - \mu_{h, w})^T) + \epsilon I$
>
> $\mathcal W_i^{mvg}(h, w)$$= \sqrt{(\phi_i(h, w) - \mu_{(h, w)})^T\Sigma_{(h, w)}^{-1}(\phi_i(h, w) - \mu_{(h, w)})}$
>
> ### Q3: The critical threshold $\tau$ should be given more explanations.
> The existence of the threshold $\tau$ is critical, while its value is not such critical because of the soft weight processing. Please find the more detailed explanations in response to Reviewer MPDh Q4. We will add these detailed explanations about $\tau$ in our manuscript.

---

### Official Review · Reviewer_P5sf · 2022-07-11

**Rating:** 7
**Confidence:** 4
**Soundness:** 3 good
**Presentation:** 3 good
**Contribution:** 4 excellent

**Summary:**

This paper sets forth a memory-based unsupervised AD method for anomaly detection. The novelty lies in efficiently denoising the data at the patch level. To this end, noise discriminators are utilized to generate outlier scores for patch-level noise elimination before corset construction. Compared to the existing algorithms, the proposed algorithm is relatively well performed.

**Questions:**

I'd like to authors to respond to the two points in the weakness.

**Limitations:**

not mentioned.

**Strengths And Weaknesses:**

Strengths:
+ A patch-level denoising strategy for coreset memory bank sounds reasonable and intersting
+ Setup a practical setting for unsupervised anomaly detection
+ Experimental results adequate

Weakness:
- No test on real noisy dataset is provided
- No computational analysis is provided

---

> ### Author Response · Authors · 2022-08-02
> **The Response to Reviewer P5sf**
>
> Thanks for your positive comments and insightful suggestions. Please find our response below.
> ### Q1: No test on a real noisy dataset is provided.
> Thank you for the suggestion. We report additional results on a real noisy dataset BTAD[1] that contains noises in the training set. In the 02 category of dataset BTAD [1], many samples with noisy small scratches greatly affect the performance of the existing algorithms. We report our results and compare them with other state-of-the-art methods on this more challenging dataset. The detailed result can be found in response to Reviewer m85i Q3. Note that our method achieves the best result on the noisy category BTAD_02 with a large margin, which demonstrates the effectiveness of our method.
>
> ### Q2: No computational analysis is provided.
> Thank you for raising an interesting point here. The computational complexity of our method is equal to PatchCore. The comparison of training and inference time is in Appendix.5 (Supplementary Material), which shows SoftCore does not require more runtime than PatchCore. We add a theoretical analysis here. The complexity of the greedy sampling in PatchCore is $\mathcal{O}(N^2h^2w^2)$, where $N$ is the number of train samples and $h,w$ is the height and weight of features. The complexity of the noise discrimination process in SoftCore-LOF is $\mathcal{O}(N^2hw)$. So the computational complexity of SoftCore is equal PatchCore by $\mathcal{O}(N^2hw+N^2h^2w^2)=\mathcal{O}(N^2h^2w^2)$. In fact, SoftCore will be faster because it removes a part of the patch as noise.
>
> References:
>
> [1] Mishra P, Verk R, Fornasier D, et al. VT-ADL: A vision transformer network for image anomaly detection and localization[C]//2021 IEEE 30th International Symposium on Industrial Electronics (ISIE). IEEE, 2021: 01-06.

---

### Official Review · Reviewer_MPDh · 2022-07-11

**Rating:** 3
**Confidence:** 5
**Soundness:** 3 good
**Presentation:** 3 good
**Contribution:** 3 good

**Summary:**

The authors propose a robust unsupervised anomaly detection model, SoftCore, that explicitly detects outliers in the training data and selects an anomaly-free coreset as a memory bank.
This memory bank is used as a reference of normality to detect novel test anomalies. The training noise discriminator that detects training contamination is based on the well-known density-based method: LOF [1].
Experiments that are conducted on the MVTec dataset show that SoftCore outperforms some baselines.


References:
[1] Markus M Breunig, Hans-Peter Kriegel, Raymond T Ng, and Jörg Sander. Lof: identifying density-based local outliers. In Proceedings of the 2000 ACM SIGMOD international conference on Management of data, pages 93–104, 2000.

[2] Gong, Dong, Lingqiao Liu, Vuong Le, Budhaditya Saha, Moussa Reda Mansour, Svetha Venkatesh and Anton van den Hengel. “Memorizing Normality to Detect Anomaly: Memory-Augmented Deep Autoencoder for Unsupervised Anomaly Detection.” 2019 IEEE/CVF International Conference on Computer Vision (ICCV) (2019): 1705-1714.

[3] Liu, Tongliang and Dacheng Tao. “Classification with Noisy Labels by Importance Reweighting.” IEEE Transactions on Pattern Analysis and Machine Intelligence 38 (2016): 447-461.

[4] Alghushairy, Omar, Raed Alsini, Terence Soule and Xiaogang Ma. “A Review of Local Outlier Factor Algorithms for Outlier Detection in Big Data Streams.” Big Data Cogn. Comput. 5 (2021): 1.

[5] Pang, Guansong, Cheng Yan, Chunhua Shen, Anton van den Hengel and Xiao Bai. “Self-Trained Deep Ordinal Regression for End-to-End Video Anomaly Detection.” 2020 IEEE/CVF Conference on Computer Vision and Pattern Recognition (CVPR) (2020): 12170-12179.

[6] Liu, Boyang, Ding Wang, Kaixiang Lin, Pang-Ning Tan and Jiayu Zhou. “RCA: A Deep Collaborative Autoencoder Approach for Anomaly Detection.” IJCAI : proceedings of the conference 2021 (2021): 1505-1511 .

[7] Zhou, Chong and Randy Clinton Paffenroth. “Anomaly Detection with Robust Deep Autoencoders.” Proceedings of the 23rd ACM SIGKDD International Conference on Knowledge Discovery and Data Mining (2017): n. pag.

**Questions:**

-

**Limitations:**

The limitations of this method are not discussed

**Strengths And Weaknesses:**

The paper is not particularly well written, and it contains a relatively large number of grammatical mistakes. The main points, however, are comprehensible.
The ideas presented here are not particularly original. The idea of reweighting the training data according to their anomalousness degree has been done before (e.g. some recent references [2,3]). Using LOF for the anomaly detection of noisy data is by no means new (e.g. [4]).

Strengths
	• The comparison to other methods shows the advantage of the method when it comes to noisy training data. However, it would be beneficial to see the results on more challenging datasets.

Weaknesses:
	• The overall idea is not novel, while the specific of the methods suggested here are. The idea of reweighting the training data according to their anomalousness degree has been done before (e.g.  recent references [2,3,4]).
	• To construct the memory bank (i.e., the reference of the normality), the authors propose to reject “the top τ% patches with the highest outlier scores”. Do they implicitly assume that the ratio of the noise in the training data is known in advance? This assumption is strong in a real-world environment where the ratio of the contamination may vary according to the collected training data.
	• Experiments on other challenging state-of-the-art datasets are desirable.
	• Limitations are not thoroughly discussed (e.g., the complexity).

Clarity:
	• The paper is not particularly well written. But, the main points are comprehensible.
	• The notations of the formulas are sometimes confusing, which makes it difficult to follow. For example, in line 193, “w” denotes the anomaly weight. However, w denotes the batch position at line 140.

Relation to prior work:
The authors state that “we are the first one to study the abnormal detection with noisy data”. However, many studies have been proposed in this research field: e.g., [5,6,7]

Reproducibility:
	• The conducted experiments are well described.
	• It would be helpful if the entire code would be made accessible in case the paper is accepted.

---

> ### Author Response · Authors · 2022-08-02
> **The Response to Reviewer MPDh**
>
> Thanks for your valuable comments. We will answer the questions one by one.
> ### Q1: The idea of reweighting the noisy training data and using LOF is not new.
> Maybe there is some misunderstanding in the novelties of our work. We explored some different methods in discriminating the noisy patches, which is a part of the contribution of this work but not the main novelties located. Although we have stated in the introduction part of the manuscript, we would like to conclude the novelties of our paper and highlight them as follows:
> 1. We propose a more practical abnormal detection setting.
> 2. We propose a patch-level denoising method. This is from a motivation that an anomaly image contains not only defective areas but also large areas of normal. The patch-level denoising strategy improves the data usage rate compared to conventional sample-level denoising in the proposed task. The comparative experiments on image-level denoising and patch-level denoising in the appendix (Supplementary Material A.4 and A.5) can also illustrate the advantage of denoising at the patch level. The comparative experiments on image-level denoising and patch-level denoising in the appendix (Supplementary Material A.4) show that patch-level denoising outperforms the PatchCore with image-level denoising by 4.7%. Meanwhile, the mentioned MemAE [2] uses a memory module for reconstruction and weights for addressing, which is far different from our approach.
> 3. We propose a novel SoftCore method to classify normal and abnormal samples based on the proposed denoising strategy.
> 4. We set a baseline for unsupervised abnormal detection with noisy data, which outperforms most of the existing unsupervised AD methods under the setting of noisy data, as well as comparable to these methods under the setting without noise.
> ### Q2: There are some existing studies of anomaly detection with noisy data.
> While the papers you mentioned are relevant to our work, we note that there are key differences in both object and task. More importantly, the methods in these studies can not be applied directly to solve our problem.
> + The reference [5] deals with video anomaly where information in consecutive frames can be exploited. While our method tackle anomaly detection from a single image.
> + The papers [6, 7] focus on semantic anomaly detection in image where the anomaly is several classes in a classification dataset. For example, on MNIST dataset, 1 is abnormal in the category of 7. Unlike semantic anomaly detection, we focus on image sensory anomaly detection [8], which has recently raised much concern.
> In image sensory anomaly detection, the difference between anomaly and normal is tiny, and anomaly location is needed. Noise in image sensory anomaly detection is more similar to the normal data and brings more challenges in this setting. This is mainly used in real industrial scenarios [9], and the semantic methods have poor performance on it [10]. A more precise way to say it is "we are the first one to study the image sensory anomaly detection with noisy data." We will update the related parts in the revised version.
>
> ### Q3: Confusion about hyperparameters $\tau$.
> The hyperparameter $\tau$ is not specifically set according to the noise ratio. In fact, we use a conservative value 0.15. The experiment in Appendix.4 shows the performance of SoftCore is consistent in different noise ratios while keeping a constant threshold of $\tau$. Another piece of evidence is the ablation study of $\tau$. As shown in Figure.4, the performance is stable when the threshold $\tau$ is larger than 0.09.
> ### Q4: Limitations are not thoroughly discussed.
> Thank you for raising an interesting point here. A limitation not thoroughly discussed in the manuscript is that our algorithm may misclassify some normal features that seldom appear in the training set. As we do not have supervision information, this limitation is intractable when we regard outlier as noise. Another reviewer presents a concern about the computational limitation, and the details can be found in response to Reviewer P5sf Q2.
> ### Q5: Reproducibility
> If the paper is accepted, the codes will be released.
>
> Additional references:
>
> [8] Yang J, Zhou K, Li Y, et al. Generalized out-of-distribution detection: A survey[J]. arXiv preprint arXiv:2110.11334, 2021.
>
> [9] Tao X, Gong X, Zhang X, et al. Deep Learning for Unsupervised Anomaly Localization in Industrial Images: A Survey[J]. arXiv e-prints, 2022: arXiv: 2207.10298.
>
> [10] Bergmann P, Fauser M, Sattlegger D, et al. MVTec AD--A comprehensive real-world dataset for unsupervised anomaly detection[C]//Proceedings of the IEEE/CVF conference on computer vision and pattern recognition. 2019: 9592-9600.

---

> > ### Author Response · Authors · 2022-08-09
> > **Clarity of the rebuttal**
> >
> > Dear Reviewer MPDh,
> >
> > Since the period of discussion ends soon and we would like to be able to respond to any remaining concerns you may have, we would like to kindly ask you to let us know if there is anything we could further clarify.
> >
> > Since novelty is the main concern in your review, we want to discuss **Q2** furtherly. Except for the difference in medium, the paper [5] focuses on a different task called *end-to-end anomaly detection* which finds outliers from all input data. In contrast, the primary task of our paper is still traditional *two-step anomaly detection*. As for articles [6, 7], we would highlight that we aim for a more realistic scene, not artificial semantic anomalies and corruption noise. So we claim sensory anomaly detection with noisy data is a new paradigm.
> >
> > Best regards,
> > Authors of Paper8103

---

### Official Review · Reviewer_m85i · 2022-07-12

**Rating:** 4
**Confidence:** 5
**Soundness:** 2 fair
**Presentation:** 2 fair
**Contribution:** 1 poor

**Summary:**

1. The authors proposed to add noisy data into the training set of unsupervised anomaly detection and created a new task setting.
2. The authors introduced a denoising mechanism and investigated 3 sampling strategies for distinguishing between noisy and clean (normal) data.
3. The experimental results are superior in the new setting.


**Questions:**

1.	Explanations of the weakness mentioned are important.
2.	It is confusing in Fig.2. The explanation of notions: ’N, H, W, C’ is missing. The arrow under ‘outlier factors of all selected patches’ are misleading, which is started from all samples including noisy ones.
3.	The authors mentioned on Page 6 Line 197: ‘The soft weights, i.e., local outlier factors, have considered the local relationship around the nearest node.’ An analysis of the neighbor's influence is needed.
4.	The English statement needed to be modified. On Page3 Line 94. ‘too long a training stage limits its usage.’ is confusing.


**Strengths And Weaknesses:**

Strengths：
1.	It is interesting to create a new task that adds noisy data to normal training images for simulating real scenes.
2.	The authors made their motivation and statement clear.
3.	The authors conducted many experiments to verify their proposed methods.
Weaknesses:
1.	The novelty of the proposed method is weak. It seems to be a modified version of previous work: PatchCore, by adding a denoising process to the memory bank.
2.	The setting of ‘Overlap’ experiment is somewhat confusing and meaningless. It is unfair for methods to train and test on the same (overlap) data.
3.	In the experiments on ‘No overlap’ setting, little performance drops were seen on state-of-the-art methods, and the comparison between the proposed methods (98.6) and PatchCore (98.4) can not show the benefits of the new approach.

---

> ### Author Response · Authors · 2022-08-02
> **The Response to Reviewer m85i (Part 1)**
>
> Thanks for your valuable comments. We will answer the questions one by one.
> ### Q1: Methodology novelty is weak.
> We would highlight that this is the first work on noisy data learning in unsupervised sensory anomaly detection. As stated in our paper, noisy data learning is a more realistic paradigm for anomaly detection.
> Although our method is inspired by PatchCore, SoftCore is more practical and robust in the noisy setting with three major differences. **First**, we have a patch-level denoising module in our method. Existing denoising methods are usually implemented at the image level. But an anomaly image contains not only defective areas but also large areas of normal. The patch-level denoising strategy attempts to leverage the normal area of anomaly images, which improves the data usage rate compared to conventional sample-level denoising in the proposed task. This is one of the key motivations we build our method on the patch level. The comparative experiments on image-level denoising and patch-level denoising in the appendix (Supplementary Material A.4 and A.5) show that patch-level denoising outperforms the PatchCore with image-level denoising of 4.7%. PatchCore with denoising performs much better than the case without denoising by 25.2% in the noisy setting. Therefore, the denoising module of our method is not a trivial modification of PatchCore. **Second**, PatchCore reweights the anomaly score by searching several neighbor patches to enhance the robustness in inference. However, this robustness operation is too weak in our experiment with noisy data. Different from PatchCore, SoftCore only needs the nearest neighbor patch that is already assigned with more global confidence to calculate the anomaly score. As a result, SoftCore is more efficient and robust in the inference phase under a noisy setting. **Third**, our method explores different methods in patch-level denoising and finds LOF is a better method than nearest neighbor searching.
>
> ### Q2: The setting of the 'Overlap' experiment is meaningless and unfair.
> As the form of anomaly is uncontrollable, similar anomalies are quite likely to emerge in the testing data, especially when a system error occurs. However, there are almost no defects with the same appearance at the same position in the MVTecAD dataset. The overlap setting here is to simulate a more actual scenario where the same defects exist in several products due to system errors in production. For example, caused by the liquid leaking from a machine, certain areas of products are consistently contaminated. This is a common problem in industrial inspection. So an overlap setting is provided as a necessary reference. This setting is the same among different methods in our comparisons. In addition, we did another experiment where the overlap images were augmented in the train set to make them different from the images in the test set. The result below shows that our method still presents better robustness when the overlap samples have been transformed, though the performance of PatchCore is improved.
>
> | Setting |  Overlap with gaussian noise | Overlap with noise and blu| Overlap with rotation |  Overlap with affine transformation |
> |:-------------:|:-----------------------------:|:-----------------------------:|:---------------------:|:------------------------------------:|
> | Method | PatchCore / SoftCore(ours) | PatchCore / SoftCore(ours) | PatchCore / SoftCore(ours) | PatchCore / SoftCore(ours) |
> | Detection | 0.760 / **0.984** | 0.848 / **0.984** | 0.950 / **0.984** | 0.933 / **0.984** |
> | Localization | 0.790 / **0.969** | 0.864 / **0.970** | 0.924 / **0.978** | 0.915 / **0.978** |

---

> ### Author Response · Authors · 2022-08-02
> **The Response to Reviewer m85i (Part 2)**
>
> ### Q3: The performance advantage compared with state-of-the-art methods in the 'no overlap' setting is too small.
> As existing performance on MVTecAD dataset is already very high, the improvement space is not large from the perspective of detection accuracy. In another aspect, the small number of anomaly samples in the dataset leads to a small noise ratio, which is the reason that performance of existing methods such as PatchCore do not drop much. Even though, our method presents a clear lead (0.7) in anomaly location performance. Besides results on the generally utilized MVTecAD dataset, to demonstrate the effectiveness of our method, we report the results on a more challenging dataset, BTAD [1]. Noted that we did not add any man-made noise to this dataset. By visualizing the training samples, we find that there is already some noise(usually small scratches) in the training set of BTAD_02, which is more consistent with our setting and further demonstrates the necessity of our approach. Moreover, the BTAD_02 contains more anomaly samples with similar appearance anomalies. The result is shown below. In the category of BTAD_02, our method attains significant improvement compared to PatchCore and PaDiM. In low-noise BTAD_01 and BTAD_03, we also showed comparable performances.
>
> | category | PatchCore | PaDiM | SoftCore(ours) | Anomaly samples |
> |:--------:|:---------:|:-----:|:--------------:|:---------------:|
> | BTAD_01 | 1.000 | 1.000 | 0.999 | 50 |
> | BTAD_02 | 0.871 | 0.871 | **0.934** | 200 |
> | BTAD_03 | 0.999 | 0.971 | 0.997 | 41 |
> | Mean | 0.957 | 0.947 | **0.977** | - |
>
> ### Q4: The analysis of the neighbor's influence is needed.
> PatchCore uses a scaling anomaly score $s^*$ to account for the behavior of neighboring patches. If the nearest neighbor is an outlier in the test, PatchCore improves its robustness by searching surrounded points. However, this robustness operation is too weak in our experiment with noisy data. The soft weights in our method are from patch-level noise discrimination, which gives a higher scaling weight to the patch far from the cluster. So the soft weights have already considered the local relationship. We will add more lines to explain the influence of neighbor patches in the final version.
> ### Q5: Confusing notion and sentence.
> Thank you for pointing out the potential misunderstanding. The 'N, H, W, C' in Fig.2 donate the number, height, weight, and channels of images. Actually, the arrow under 'outlier factors of all selected patches' starts from the module of the whole green block. We will add more illustrations of these notations and modify the mistakes in figures to reduce misunderstanding.
> 'too long a training stage limits its usage.' means the training stage that takes too much time limits the usage of the knowledge distillation method. For the confusing sentence, we will improve the English usage.
> Some of the above experiments and statements will be added to the revised version of the paper.
>
> References:
>
> [1] Mishra P, Verk R, Fornasier D, et al. VT-ADL: A vision transformer network for image anomaly detection and localization[C]//2021 IEEE 30th International Symposium on Industrial Electronics (ISIE). IEEE, 2021: 01-06.

---

> > ### Author Response · Authors · 2022-08-09
> > **Clarity of the answer**
> >
> > Dear Reviewer m85i,
> >
> > Since the period of discussion ends soon and we would like to be able to respond to any remaining concerns you may have, we would like to kindly ask you to let us know if there is anything we could further clarify.
> >
> > Best regards,
> > Authors of Paper8103

---

### Meta-Review · Area_Chair_NeYY · 2022-08-24

**Recommendation:** Accept
**Confidence:** Less certain

**Metareview:**

The paper proposes a memory-based unsupervised anomaly detection approach that efficiently denoises the data at the patch level. This paper has quite diverse evaluations. Some reviewers are concerned that the novelty of the proposed approach is low, and the paper conducts inappropriate experiments. On the other hand, some reviewers appreciate that the proposed approach is reasonable and solves a practical problem. Since the paper could be a pioneering work in the field, I recommend to accept the paper.

**Award:**

No

---

### Meta-Review · Area_Chair_NeYY · 2022-09-14

**Recommendation:** Accept
**Confidence:** Less certain

**Metareview:**

Can the authors please remove the word "SoftCore" from the title and manuscript as this word has negative connotations.

**Award:**

No

---

### Decision · Program_Chairs · 2022-09-14

Accept